# Multi-Modal Knowledge Graph Transformer Framework for Multi-Modal Entity Alignment

**Qian Li[1,2], Cheng Ji[1,2], Shu Guo[3], Zhaoji Liang[4], Lihong Wang[3], Jianxin Li[1,2]\***

[1]School of Computer Science and Engineering, Beihang University, Beijing, China
[2]Beijing Advanced Innovation Center for Big Data and Brain Computing, Beijing, China
[3]National Computer Network Emergency Response Technical Team/Coordination Center of China
[4]School of Computer Science and Technology, University of Chinese Academy of Sciences
{liqian, jicheng, lijx}@act.buaa.edu.cn, guoshu@cert.org.cn,
liangzhaoji23@mails.ucas.ac.cn, wlh@isc.org.cn

## Abstract

Multi-Modal Entity Alignment (MMEA) is a critical task that aims to identify equivalent entity pairs across multi-modal knowledge graphs (MMKGs). However, this task faces challenges due to the presence of different types of information, including neighboring entities, multi-modal attributes, and entity types. Directly incorporating the above information (*e.g.*, concatenation or attention) can lead to an unaligned information space. To address these challenges, we propose a novel MMEA transformer, called MoAlign, that hierarchically introduces neighbor features, multi-modal attributes, and entity types to enhance the alignment task. Taking advantage of the transformer's ability to better integrate multiple information, we design a hierarchical modifiable self-attention block in a transformer encoder to preserve the unique semantics of different information. Furthermore, we design two entity-type prefix injection methods to integrate entity-type information using type prefixes, which help to restrict the global information of entities not present in the MMKGs. Our extensive experiments on benchmark datasets demonstrate that our approach outperforms strong competitors and achieves excellent entity alignment performance.

## 1 Introduction

Multi-modal entity alignment (MMEA) is a challenging task that aims to identify equivalent entity pairs across multiple knowledge graphs that feature different modalities of attributes, such as text and images. To accomplish this task, sophisticated models are required to effectively leverage information from different modalities and accurately align entities. This task is essential for various applications, such as cross-lingual information retrieval, question answering (Antol et al., 2015; Shih et al., 2016), and recommendation systems (Sun et al., 2020; Xu et al., 2021).

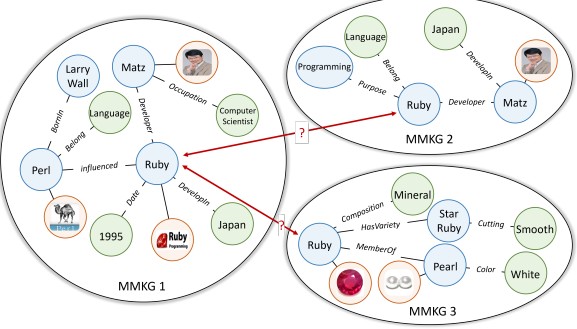

Figure 1: Two examples of the MMEA task, where the entity pair in MMKG 1 and MMKG 2 is entity seed and the pair in MMKG 1 and MMKG 3 is not. The blue, green, and orange circles are entities, textual attributes and visual attributes.

MMEA (Liu et al., 2019; Li et al., 2023b; Liu et al., 2021; Lin et al., 2022) is challenging due to the heterogeneity of MMKGs (*e.g.*, different neighbors, multi-modal attributes, distinct types), which makes it difficult to learn rich knowledge representations. Previous approaches such as PoE (Liu et al., 2019) concatenated all modality features to create composite entity representations but failed to capture interactions among heterogeneous modalities. More recent works (Chen et al., 2020; Guo et al., 2021) designed multi-modal fusion modules to better integrate attributes and entities, but still did not fully exploit the potential interactions among modalities. These methods also ignored inter-modality dependencies between entity pairs, which could lead to incorrect alignment. Generally speaking, although MMKGs offer rich attributes and neighboring entities that could be useful for multi-mdoal entity alignment, current methods have limitations in (i) ignoring the differentiation and personalization of the aggregation of heterogeneous neighbors and modalities leading to the misalignment of cross-modal semantics, and (ii) lacking the use of entity heterogeneity resulting in the non-discriminative representations of different

---
\*Corresponding author.

meaning/types of entities.

Therefore, the major challenge of MMEA task is how to perform differentiated and personalized aggregation of heterogeneous information of the neighbors, modalities, and types. Although such information is beneficial to entity alignment, directly fusing will lead to misalignment of the information space, as illustrated in Figure 1. Firstly, notable disparities between different modalities make direct alignment a challenging task. For example, both the visual attribute of entity *Ruby* in MMKG1 and the neighbor information of the entity *Ruby* in MMKG2 contain similar semantics of *programming*, but data heterogeneity may impede effective utilization of this information. Secondly, complex relationships between entities require a thorough understanding and modeling of contextual information and semantic associations. Entities such as the *Ruby*, the *Perl*, and the entity *Larry Wall* possess unique attributes, and their inter-relationships are non-trivial, necessitating accurate modeling based on contextual information and semantic associations. Furthermore, the existence of multiple meanings for entities further exacerbates the challenge of distinguishing between two entities, such as in the case of the *Ruby*, which has different meanings in the MMKG1 and MMKG3 where it may be categorized as a jewelry entity or a programming language entity, respectively.

To overcome the aforementioned challenges, we propose a novel Multi-Modal Entity Alignment Transformer named MoAlign[1]. Our framework hierarchically introduces neighbor, multimodal attribute, and entity types to enhance the alignment task. We leverage the transformer architecture, which is known for its ability to process heterogeneous data, to handle this complex task. Moreover, to enable targeted learning on different modalities, we design a hierarchical modifiable self-attention block in the Transformer encoder, which builds associations of task-related intra-modal features through the layered introduction. Additionally, we introduce positional encoding to model entity representation from both structure and semantics simultaneously. Furthermore, we integrate entity-type information using an entity-type prefix, which helps to restrict the global information of entities that are not present in the multi-modal knowledge graphs. This prefix enables better filtering out of

---

[1] The source code is available at https://github.com/xiaoqian19940510/MoAlign.

unsuitable candidates and further enriches entity representations. To comprehensively evaluate the effectiveness of our proposed approach, we design training objectives for both entity and context evaluation. Our extensive experiments on benchmark datasets demonstrate that our approach outperforms strong competitors and achieves excellent entity alignment performance. Our contributions can be summarized as follows.

- We propose a novel MMEA framework named MoAlign, which effectively integrates heterogeneous information through the multi-modal KG Transformer.

- We design a hierarchical modifiable self-attention block to build associations of task-related intra-modal features through the layered introduction and design an entity-type prefix to further enrich entity representations.

- Experimental results indicate that the framework achieves state-of-the-art performance on the public multi-modal entity alignment datasets.

## 2 Preliminaries

**Multi-Modal Knowledge Graph.** A multi-modal knowledge graph (MKG) is represented by four sets: entities ($\mathcal{E}$), relations ($\mathcal{R}$), multi-modal attributes ($\mathcal{A}$), and triplets ($\mathcal{T}$). The size of each set is denoted by $N_{\mathcal{E}}$, $N_{\mathcal{R}}$, and $N_{\mathcal{A}}$. The multi-modal attributes are divided into text ($\mathcal{A}T$) and image ($\mathcal{A}I$) attributes. The relation set $\mathcal{R}$ includes entity relations ($\mathcal{R}\mathcal{E}$), text attribute relations ($\mathcal{R}T$), and image attribute relations ($\mathcal{R}I$). The set of triplets $\mathcal{T}$ includes entity triplets, text attribute triplets, and image attribute triplets.

**Multi-Modal Entity Alignment Task.** Multi-modal entity alignment (Chen et al., 2020; Guo et al., 2021; Liu et al., 2021; Chen et al., 2022) aims to determine if two entities from different multi-modal knowledge graphs refer to the same real-world entity. This involves calculating the similarity between pairs of entities, known as alignment seeds. The goal is to learn entity representations from two multi-modal knowledge graphs ($MKG_1$ and $MKG_2$) and calculate the similarity between a pair of entity alignment seeds $(e, e')$ taken from these KGs. The set of entity alignment seeds is denoted as $\mathcal{S} = \{(e, e') \mid e \in \mathcal{E}, e' \in \mathcal{E}', e \equiv e'\}$.

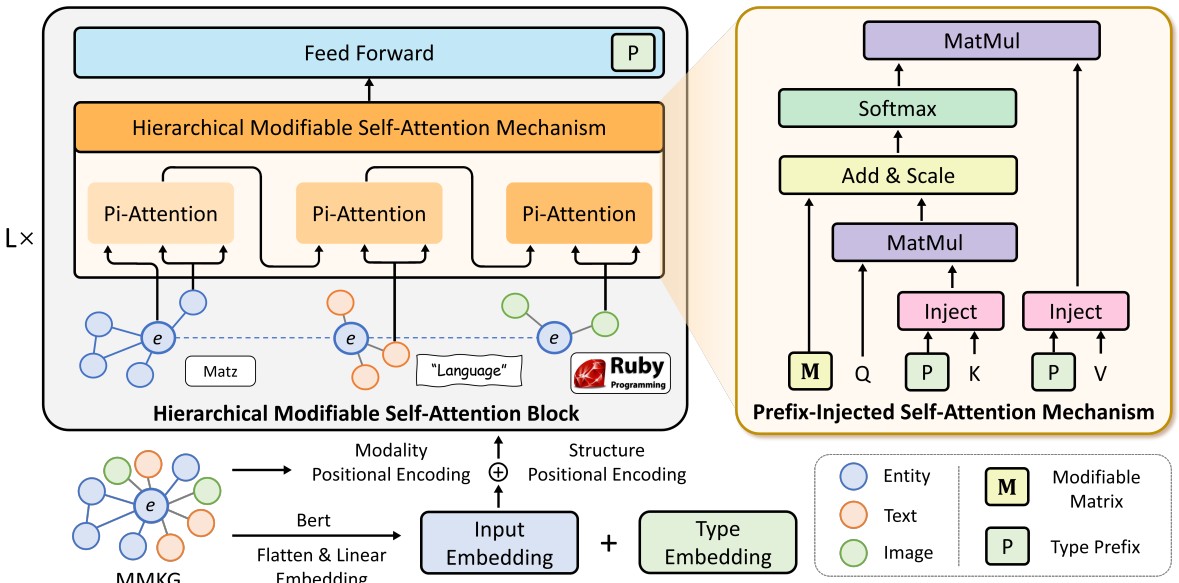

Figure 2: The framework of the Multi-Modal KG Transformer, MoAlign. The hierarchical modifiable self-attention block learns the entity by hierarchically integrating multi-modal attributes and neighbors. The prefix-injected self-attention mechanism introduces entity type information for alignment.

## 3 Framework

This section introduces our proposed framework MoAlign. As shown in Figure 2, we introduce positional encoding to simultaneously model entity representation from both modality and structure. To hierarchically introduce neighbor and multi-modal attributes, we design a hierarchical modifiable self-attention block. This block builds associations of task-related intra-modal features through the layered introduction. Furthermore, for integrating entity-type information, we design a prefix-injected self-attention mechanism, which helps to restrict the global information of entities not present in the MMKGs. Additionally, MoAlign also design training objectives for both entity and context evaluation to comprehensively assess the effectiveness.

### 3.1 Multi-Modal Input Embedding

#### 3.1.1 Multi-Modality Initialization

The textual attribute is initialized by BERT (Devlin et al., 2019). For the visual attributes, to enable direct processing of images by a standard transformer, the image is split into a sequence of patches (Dosovitskiy et al., 2021). We then perform a flatten operation to convert the matrix into a one-dimensional vector similar to word embeddings, which is similar to embeddings in BERT (Devlin et al., 2019) and concatenate them to form the image embedding vector, denoted as $\boldsymbol{v}_v$. However, the initialization

of word embeddings follows a specific distribution, whereas the distribution of image embeddings generated using this method is not consistent with the pretraining model's initial distribution. Therefore, to standardize the distribution of image embeddings to be consistent with the distribution of the vocabulary used by the pretraining model, we perform the following operation:

$$\boldsymbol{v}_v \leftarrow (\boldsymbol{v}_v - mean(\boldsymbol{v}_v))/std(\boldsymbol{v}_v) \cdot \lambda, \quad (1)$$

where $mean(\boldsymbol{v}_v)$ and $std(\boldsymbol{v}_v)$ are the mean and standard deviation of $\boldsymbol{v}_v$, respectively, and $\lambda$ is the standard deviation of truncated normal function.

#### 3.1.2 Positional Encoding

For the transformer input, we additionally input two types of positional encoding to maintain structure information of multi-modal KG as follows.

**Modality Positional Encoding.** To enable the model to effectively distinguish between entities, textual attributes, image attributes, and introduced entity types, we incorporate a unique position code for each modality. These position codes are then passed through the encoding layers of the model to enable it to differentiate between different modalities more accurately and learn their respective features more effectively. Specifically, we assign position codes of 1, 2, 3, and 4 to entities, textual attributes, image attributes, and introduced entity

types, respectively. By incorporating this modality positional encoding information into the encoding process, our proposed model is able to incorporate modality-specific features into the alignment task.

**Structure Positional Encoding.** To capture the positional information of neighbor nodes, we introduce a structure positional encoding that assigns a unique position code to each neighbor. This allows the model to distinguish between them and effectively incorporate the structure information of the knowledge graph into alignment process. Specifically, we assign a position code of 1 to the current entity and its attributes. For first-order neighbors, we randomly initialize a reference order and use it to assign position codes to each neighbor and its corresponding relation as $2n$ and $2n + 1$, respectively. Additionally, we assign the same structure positional encoding of attributes to their corresponding entities. By doing so, the model can differentiate between different neighbor nodes and effectively capture their positional information.

To fully utilize the available information, we extract relation triplets and multi-modal attribute triplets for each entity. The entity is formed by combining multi-modal sequences as $\{e,(e_1,r_1),\ldots,(e_n,r_n),(a_1,v_1),\ldots,(a_m,v_m),e_T\}$, where $(e_i,r_i)$ represents the $i$-th neighbor of entity $e$ and its relation. $(a_j,v_j)$ represents the $j$-th attribute of entity $e$ and its value $v_j$, and it contains textual and visual attributes. $n$ and $m$ are the numbers of neighbors and attributes. $e_T$ is the type embeddding.

## 3.2 Hierarchical Modifiable Self-Attention

To better capture the dependencies and relationships of entities and multi-modal attributes, we incorporate cross-modal alignment knowledge into the transformer. Specifically, a hierarchical modifiable self-attention block is proposed to better capture complex interactions between modalities and contextual information. The block focuses on associations of inter-modal (between modalities, such as textual and visual features) by cross-modal attention. By doing so, the model can capture more informative and discriminative features.

### 3.2.1 Hierarchical Multi-Head Attention

We propose a novel hierarchical block that incorporates distinct attention mechanisms for different inputs, which facilitates selective attention toward various modalities based on their significance in the alignment task. The transformer encoder comprises three separate attention mechanisms, namely neighbor attention, textual attention, and visual attention. We utilize different sets of queries, keys, and values to learn diverse types of correlations between the input entity, its attributes, and neighbors.

**Neighbor Multi-Head Attention.** More specifically, in the first layer of the hierarchical block of the $l$-th transformer layer, we employ the input entity as queries and its neighbors as keys and values to learn relations between the entity and its neighbors:

$$\boldsymbol{e}^{(l.1)} = \text{MH-Attn}(\boldsymbol{e}, \boldsymbol{e}_i, \boldsymbol{e}_i), \qquad (2)$$

where $\boldsymbol{e}_i$ is the neighbor of entity $e$ and $\boldsymbol{e}^{(l.1)}$ is the representation of entity $e$ in the first layer.

**Textual Multi-Head Attention.** In the second layer of the hierarchical block, the input entity and its textual attributes are used as queries and keys/values to learn the correlations between the entity and its textual attributes.

$$\boldsymbol{e}^{(l.2)} = \text{MH-Attn}(\boldsymbol{e}^{(l.1)}, [\boldsymbol{a}_t; \boldsymbol{v}_t], [\boldsymbol{a}_t; \boldsymbol{v}_t]), \quad (3)$$

where $(\boldsymbol{a}_t, \boldsymbol{v}_t)$ represents the textual attribute of entity $e$ and its value, and $\boldsymbol{e}^{(l.2)}$ is the representation of entity $e$ in the second layer.

**Visual Multi-Head Attention.** In the third layer of the hierarchical block, the input entity and its visual attributes are used similarly to the second layer, to learn the correlations between the entity and its visual attributes.

$$\boldsymbol{e}^{(l.3)} = \text{MH-Attn}(\boldsymbol{e}^{(l.2)}, [\boldsymbol{a}_v; \boldsymbol{v}_v], [\boldsymbol{a}_v; \boldsymbol{v}_v]), \quad (4)$$

where $(\boldsymbol{a}_v, \boldsymbol{v}_v)$ represents the visual attribute of entity $e$ and its value, and $\boldsymbol{e}^{(l.3)}$ is the representation of entity $e$ in the third layer. By incorporating neighbor attention, textual attribute attention, and visual attribute attention, our model can capture various correlations between the input entity, its attributes, and neighbors in a more effective manner.

### 3.2.2 Modifiable Self-Attention

To learn a specific attention matrix for building correlations among entities and attributes, we design a modifiable self-attention mechanism. We manage to automatically and adaptively generate an information fusion mask matrix based on sequence features. The length of the sequence and the types of information it contains may vary depending on the entity, as some entities may lack certain attribute information and images.

**Modifiable Self-Attention Mechanism.** To adapt to the characteristics of different sequences, it is necessary to assign "labels" to various information when generating the sequence, such as using $[E]$ to represent entities and $[R]$ to represent relations. This way, the positions of various information in the sequence can be generated, and the mask matrix can be generated accordingly.

These labels need to be processed by the model's tokenizer after inputting the model and generating the mask matrix to avoid affecting the subsequent generation of embeddings. We can still modify the vocabulary to allow the tokenizer to recognize these words and remove them. The mask matrix can be generated based on the positions of various information, represented by labels, as follows:

$$\mathbf{M}_{ij} = \begin{cases} 1, & \text{if}(i,j,\cdot) \in \mathcal{T} \text{ or } T(i) = T(j) \\ 0, & \text{otherwise} \end{cases}, \quad (5)$$

where $T(\cdot)$ is the type mapping function of entities and attributes.

Subsequently, residual connections and layer normalization are utilized to merge the output of hierarchical modifiable self-attention $e^l$ and apply a position-wise feed-forward network to each element in the output sequence of the self-attention. The output sequence, residual connections, and layer normalization are employed as:

$$e^{(l)} = \text{LayerNorm}\left(e^{(l-1)} + \text{FFN}(e^{(l)})\right), \quad (6)$$

where $e^l$ is the output of the hierarchical modifiable self-attention in the $l$-th layer of the transformer. The use of residual connections and layer normalization helps stabilize training and improve performance. It enables our model to better understand and attend to different modalities, leading to more accurate alignment results.

### 3.3 Entity-Type Prefix Injection

Prefix methods can help improve alignment accuracy by providing targeted prompts to the model to improve generalization, including entity type information (Liu et al., 2023). The entity-type injection introduces a type of information that takes as input a pair of aligned seed entities from different MMKGs and uses entities as prompts. The goal is to inject entity-type information into the multi-head attention and feed-forward neural networks.

**Prefix-Injected Self-Attention Mechanism.** Specifically, we create two sets of prefix vectors for entities, $\boldsymbol{p}^k, \boldsymbol{p}^v \in \mathbb{R}^{n_t \times d}$, for keys and values separately, where $n_t$ is the number of entity types. These prefix vectors are then concatenated with the original key $K$ and value $V$. The multi-head attention in Eq.(2) is performed on the newly formed prefixed keys and values as follows:

$$\begin{aligned} \text{MH-Attn}(\mathbf{Q},\mathbf{K},\mathbf{V}) &= \text{Concat}(\text{h}_1,\cdots,\text{h}_N)\mathbf{W}_o, \\ \text{h}_i &= \text{Pi-Attn}\left(\mathbf{Q}\mathbf{W}_i^q, [\mathbf{K}\mathbf{W}_i^k; \boldsymbol{p}^k], [\mathbf{V}\mathbf{W}_i^v; \boldsymbol{p}^v]\right), \end{aligned} \quad (7)$$

where $d_N$ is typically set to $d/N$, and $N$ is number of head. $\text{h}_i$ denotes $i$-th output of self-attention in $l$-th layer. $\mathbf{W}_o \in \mathbb{R}^{d \times d}$, $\mathbf{W}_i^q, \mathbf{W}_i^k, \mathbf{W}_i^v \in \mathbb{R}^{d \times d_N}$ are learnable parameters. In this paper, we use the type embedding $\boldsymbol{e}_T$ as the prefix.

In order to effectively incorporate alignment information, we utilize the mask token to influence attention weights. The mask token serves as a means to capture effective neighbors and reduce computational complexity. To achieve this, we set attention weights $m$ to a large negative value based on the mask matrix $\mathbf{M}_{ij}$ in the modifiable self-attention mechanism. Consequently, attention weights for the mask token are modified as follows:

$$\text{Pi-Attn}(\mathbf{Q},\mathbf{K}_t,\mathbf{V}_t) = \text{Softmax}\left(\frac{\mathbf{Q}\mathbf{K}_t^{\text{T}} + m}{\sqrt{d_k}}\right)\mathbf{V}_t, \quad (8)$$

where $\mathbf{Q}$, $\mathbf{K}_t$, and $\mathbf{V}_t$ represent the query of the entity, the key containing entity type, and the value containing entity type matrices, respectively. $d_k$ refers to the dimensionality of key vectors, while $m$ represents the attention weights.

**Prefix-Injected Feed Forward Network.** Recent research suggests that the feed-forward layers within the transformer architecture store factual knowledge and can be regarded as unnormalized key-value memories (Yao et al., 2022). Inspired by this, we attempt to inject entity type into each feed-forward layer to enhance the model's ability to capture the specific information related to the entity type. Similar to prefix injection in attention, we first repeat the entity type $\boldsymbol{e}_T$ to create two sets of vectors, $\boldsymbol{\Phi}^k, \boldsymbol{\Phi}^v \in \mathbb{R}^{n_t \times d}$. These vectors are then concatenated with the original parameter matrices of the first and second linear layers.

$$\text{FFN}(\mathbf{E}) = f\left(\mathbf{E} \cdot [\mathbf{W}_f^k; \boldsymbol{\Phi}^k]\right) \cdot [\mathbf{W}_f^v; \boldsymbol{\Phi}^v], \quad (9)$$

where $f$ denotes the non-linearity function, $\mathbf{E}$ represents the output of the hierarchical modifiable self-attention. $\mathbf{W}_f^k, \mathbf{W}_f^v$ are the parameters.

## 3.4 Training Objective

In order to effectively evaluate the MMEA from multiple perspectives, we propose a training objective function that incorporates both aligned entity similarity and context similarity. This objective function serves to assess the quality of entity alignment and takes into consideration both the entities themselves and their surrounding context.

**Aligned Entity Similarity.** The aligned entity similarity constraint loss is designed to measure similarity between aligned entity pairs and ensure that embeddings of these entities are close to each other in the embedding space.

$$\mathcal{L}_{\text{EA}} = \text{sim}(\boldsymbol{e}, \boldsymbol{e}') - \text{sim}(\boldsymbol{e}, \overline{\boldsymbol{e}}') - \text{sim}(\overline{\boldsymbol{e}}, \boldsymbol{e}'), \quad (10)$$

where $(\boldsymbol{e}, \boldsymbol{e}')$ represent the final embeddings of the aligned seed entities $(e, e')$ from knowledge graphs $KG1$ and $KG2$. $\overline{\boldsymbol{e}}$ and $\overline{\boldsymbol{e}}'$ denote the negative samples of the seed entities. $\text{sim}(\cdot, \cdot)$ refers to the cosine distance between the embeddings.

**Context Similarity.** The context similarity loss is employed to ensure that the context representations of aligned entities are close to each other in the embedding space.

$$\mathcal{L}_{\text{Con}} = \text{sim}(\boldsymbol{o}, \boldsymbol{o}') - \text{sim}(\boldsymbol{o}, \overline{\boldsymbol{o}}') - \text{sim}(\overline{\boldsymbol{o}}, \boldsymbol{o}'), \quad (11)$$

where $(\boldsymbol{o}, \boldsymbol{o}')$ denote the final context representations ([cls] embedding) of the aligned seed entities $(e, e')$. $\overline{\boldsymbol{o}}$ and $\overline{\boldsymbol{o}}'$ represent the negative samples of the seed entities' context.

The total alignment loss $\mathcal{L}$ is computed as:

$$\mathcal{L} = \alpha \mathcal{L}_{\text{EA}} + \beta \mathcal{L}_{\text{Con}}, \quad (12)$$

where $\alpha, \beta$ are learnable hyper-parameters.

## 4 Experiment

### 4.1 Dataset

We have conducted experiments on two of the most popular datasets, namely FB15K-DB15K and FB15K-YAGO15K, as described in (Liu et al., 2019). The FB15K-DB15K dataset[2] is an entity alignment dataset of FB15K and DB15K MMKGs, while the latter is a dataset of FB15K and YAGO15K MMKGs. Consistent with prior works (Chen et al., 2020; Guo et al., 2021), we split each dataset into training and testing sets in proportions of 2:8, 5:5, and 8:2, respectively. The MRR, Hits@1, and Hits@10 are reported for evaluation on different proportions of alignment seeds.

[2]https://github.com/mniepert/mmkb

## 4.2 Comparision Methods

We compare our method with three EA baselines (**TransE** (Bordes et al., 2013), **GCN-align** (Wang et al., 2018a), and **AttrGNN** (Liu et al., 2020b)), which aggregate text attributes and relation information, and introduced image attributes initialized by VGG16 for entity representation using the same aggregation method as for text attributes. We further use three transformer models (**BERT** (Devlin et al., 2019), **ViT** (Dosovitskiy et al., 2021), and **CLIP** (Radford et al., 2021)) to incorporate multi-modal information, and three MMEA methods (**PoE** (Liu et al., 2019), **Chen et al.** (Chen et al., 2020), **HEA** (Guo et al., 2021), **EVA** (Liu et al., 2021), and **ACK-MMEA** (Li et al., 2023a)) to focusing on utilizing multi-modal attributes. The detail is given in Appendix B.

### 4.3 Implementation Details

We utilized the optimal hyperparameters reported in the literature for all baseline models. Our model was implemented using PyTorch, an open-source deep learning framework. We initialized text and image attributes using bert-base-uncased[3] and VGG16[4], respectively. To ensure fairness, all baselines were trained on the same data set partition. The best random dropping rate is 0.35, and coefficients $\alpha, \beta$ were set to 5 and 2, respectively. All hyperparameters were tuned on validation data using 5 trials. All experiments were performed on a server with one GPU (Tesla V100).

### 4.4 Main Results

To verify the effectiveness of MoAlign, we report overall average results in Table 1. It shows performance comparisons on both two datasets with different splits on training/testing data of alignment seeds, *i.e.*, 2:8, 5:5, and 8:2. From the table, we can observe that: 1) Our model outperforms all baselines of both EA, multi-modal Transformer-based, and MMEA methods, in terms of three metrics on both datasets. It demonstrates that our model is robust to different proportions of training resources and learns a good performance on few-shot data. 2) Compared to EA baselines (1-3), especially for MRR and Hits@1, our model improves 5% and 9% up on average on FB15K-DB15K and FB15K-YAGO15K, tending to achieve more significant improvements. It demonstrates that the effective-

[3]https://github.com/huggingface/transformers
[4]https://github.com/machrisaa/tensorflow-vgg

**Table 1** (top): FB15K-DB15K

| Methods | FB15K-DB15K (20%) | | | FB15K-DB15K (50%) | | | FB15K-DB15K (80%) | | |
|---|---|---|---|---|---|---|---|---|---|
| | MRR | Hits@1 | Hits@10 | MRR | Hits@1 | Hits@10 | MRR | Hits@1 | Hits@10 |
| TransE (Bordes et al., 2013) | 13.4 | 7.8 | 24.0 | 30.6 | 23.0 | 44.6 | 50.7 | 42.6 | 65.9 |
| GCN-align (Wang et al., 2018a) | 8.7 | 5.3 | 17.4 | 29.3 | 22.6 | 43.5 | 47.2 | 41.4 | 63.5 |
| AttrGNN (Liu et al., 2020b) | 34.3 | 25.2 | 53.5 | 54.7 | 47.3 | 72.1 | 70.3 | 67.1 | 83.9 |
| BERT (Devlin et al., 2019) | 32.6 | 24.3 | 48.0 | 49.6 | 45.2 | 67.9 | 65.3 | 64.5 | 80.1 |
| ViT (Dosovitskiy et al., 2021) | 33.5 | 25.1 | 53.9 | 50.5 | 45.5 | 69.0 | 71.5 | 66.8 | 85.7 |
| CLIP (Radford et al., 2021) | 35.4 | 27.0 | 55.3 | 54.1 | 48.7 | 71.4 | 73.9 | 68.3 | 86.0 |
| PoE (Liu et al., 2019) | 17.0 | 12.6 | 25.1 | 53.3 | 46.4 | 65.8 | 72.1 | 66.6 | 82.0 |
| Chen et al. (Chen et al., 2020) | 35.7 | 26.5 | 54.1 | 51.2 | 41.7 | 70.3 | 68.5 | 59.0 | 86.9 |
| HEA (Guo et al., 2021) | - | 12.7 | 36.9 | - | 26.2 | 58.1 | - | 41.7 | 78.6 |
| EVA (Liu et al., 2021) | 35.2 | 28.9 | 54.5 | 53.8 | 45.3 | 72.9 | 71.6 | 63.5 | 85.1 |
| ACK-MMEA (Li et al., 2023a) | 38.7 | 30.4 | 54.9 | 62.4 | 56.0 | 73.6 | 75.2 | 68.2 | 87.4 |
| **MoAlign (ours)** | **40.9** (↑2.2) | **31.8** (↑1.4) | **56.4** (↑1.1) | **63.4** (↑1.0) | **57.6** (↑1.6) | **74.9** (↑1.3) | **77.3** (↑2.1) | **69.9** (↑1.6) | **88.2** (↑0.8) |

**Table 1** (bottom): FB15K-YAGO15K

| Methods | FB15K-YAGO15K (20%) | | | FB15K-YAGO15K (50%) | | | FB15K-YAGO15K (80%) | | |
|---|---|---|---|---|---|---|---|---|---|
| | MRR | Hits@1 | Hits@10 | MRR | Hits@1 | Hits@10 | MRR | Hits@1 | Hits@10 |
| TransE (Bordes et al., 2013) | 11.2 | 6.4 | 20.3 | 26.2 | 19.7 | 38.2 | 46.3 | 39.2 | 59.5 |
| GCN-align (Wang et al., 2018a) | 15.3 | 8.1 | 23.5 | 29.4 | 23.5 | 42.4 | 47.7 | 40.6 | 64.3 |
| AttrGNN (Liu et al., 2020b) | 31.8 | 22.4 | 39.5 | 46.2 | 38.0 | 63.9 | 67.1 | 59.9 | 78.7 |
| BERT (Devlin et al., 2019) | 30.5 | 23.6 | 39.0 | 48.7 | 43.1 | 62.4 | 67.3 | 60.8 | 81.2 |
| ViT (Dosovitskiy et al., 2021) | 32.4 | 26.8 | 44.9 | 52.0 | 45.7 | 67.5 | 71.3 | 63.1 | 82.0 |
| CLIP (Radford et al., 2021) | 34.8 | 29.3 | 47.1 | 56.8 | 49.0 | 70.2 | 72.1 | 65.2 | 85.2 |
| PoE (Liu et al., 2019) | 15.4 | 11.3 | 22.9 | 41.4 | 34.7 | 53.6 | 63.5 | 57.3 | 74.6 |
| Chen et al. (Chen et al., 2020) | 31.7 | 23.4 | 48.0 | 48.6 | 40.3 | 64.5 | 68.2 | 59.8 | 83.9 |
| HEA (Guo et al., 2021) | - | 10.5 | 31.3 | - | 26.5 | 58.1 | - | 43.3 | 80.1 |
| EVA (Liu et al., 2021) | 33.5 | 25.0 | 46.2 | 56.1 | 47.8 | 68.3 | 72.5 | 64.0 | 84.5 |
| ACK-MMEA (Li et al., 2023a) | 36.0 | 28.9 | 49.6 | 59.3 | 53.5 | 69.9 | 74.4 | 67.6 | 86.4 |
| **MoAlign (ours)** | **37.8** (↑1.8) | **29.6** (↑0.3) | **52.5** (↑2.9) | **61.7** (↑2.4) | **55.0** (↑1.5) | **71.3** (↑1.1) | **76.9** (↑2.5) | **68.9** (↑1.3) | **88.4** (↑2.0) |

Table 1: Main experiments on FB15K-DB15K (top) (%) and FB15K-YAGO15K (bottom) (%) with different proportions of MMEA seeds. The best results are highlighted in bold, and underlined values are the second best result. "↑" means the improvement compared to the second best result, and "-" means that results are not available.

| Variants | FB15K-DB15K (80%) | | | | FB15K-DB15K (80%) | | | |
|---|---|---|---|---|---|---|---|---|
| | MRR | Hits@1 | Hits@10 | △ Avg | MRR | Hits@1 | Hits@10 | △ Avg |
| **MoAlign (ours)** | **77.3** | **69.9** | **88.2** | - | **76.9** | **68.9** | **88.4** | - |
| w/o modifiable layer | 75.8 | 66.4 | 85.9 | ↓2.4 | 74.2 | 65.3 | 85.7 | ↓3.0 |
| w/o modifiable self-attention | 76.5 | 67.1 | 86.4 | ↓1.8 | 74.6 | 66.7 | 86.2 | ↓2.2 |
| repl. multi-head self-attention | 76.0 | 66.9 | 87.1 | ↓1.8 | 74.5 | 65.9 | 86.0 | ↓2.6 |
| w/o type information | 76.2 | 67.8 | 87.3 | ↓1.4 | 74.1 | 66.5 | 86.7 | ↓2.3 |
| w/o context loss | 76.9 | 68.2 | 87.1 | ↓1.1 | 75.4 | 67.2 | 86.3 | ↓1.8 |
| w/o text attribute | 75.1 | 65.3 | 85.7 | ↓3.1 | 73.4 | 65.0 | 86.4 | ↓3.1 |
| w/o image attribute | 75.8 | 65.4 | 86.0 | ↓2.7 | 74.0 | 66.8 | 85.4 | ↓2.7 |

Table 2: Variant experiments on FB15K-DB15K (80%) and FB15K-YOGA15K (80%). "w/o" means removing corresponding module from complete model. "repl." means replacing corresponding module with other module.

ness of multi-modal context information benefits incorporating alignment knowledge. 3) Compared to multi-modal transformer-based baselines, our model achieves better results and the transformer-based baselines perform better than EA baselines. It demonstrates that transformer-based structures can learn better MMEA knowledge. 4) Compared to MMEA baselines, our model designs a Hierarchical Block and modifiable self-attention mechanism, the average gains of our model regarding MRR, Hits@1, and Hits@10 are 2%, 1.4%, and 1.7%, respectively. The reason is that our method incorporates multi-modal attributes and robust context entity information. All the observations demonstrate the effectiveness of MoAlign.

## 4.5 Discussions for Model Variants

To investigate the effectiveness of each module in MoAlign, we conduct variant experiments, showcasing the results in Table 2. The "↓" means the value of performance degradation compared to the MoAlign. We can observe that: 1) The impact of the Hierarchical Block tends to be more significant. We believe the reason is that the adaptive introduction of multi-modal attributes and neighbors captures more clues. 2) By replacing the modifiable self-attention to the multi-head self-attention, the performance decreased significantly. It demonstrates that the modifiable self-attention captures more effective multi-modal attribute and relation information. 3) When we remove all image at-

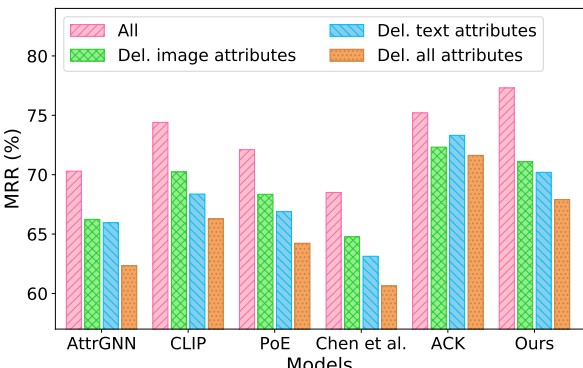

Figure 3: Results of deleting attributes on FB15K-DB15K (80%). "Del." means deleting the corresponding attribute.

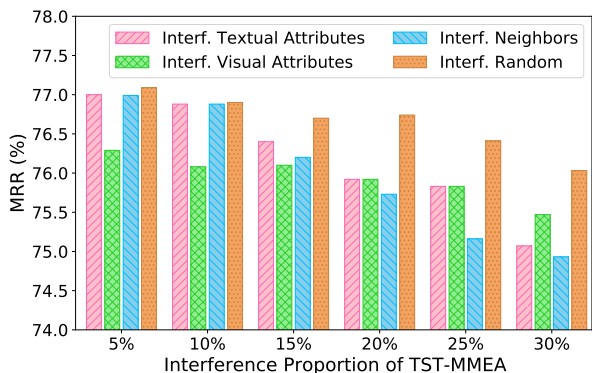

Figure 4: Results of interference attributes or neighbors on FB15K-DB15K (80%). "Interf." means interference some proportion of attributes or neighbors.

| Variants | FB15K-DB15K (80%) | | | |
|---|---|---|---|---|
| | MRR | Hits@1 | Hits@10 | △ Avg |
| **MoAlign (N → V → T)** | **77.3** | **69.9** | **88.2** | - |
| MoAlign (N → T → V) | 76.3 | 68.0 | 87.5 | ↓1.2 |
| MoAlign (V → N → T) | 76.0 | 67.4 | 86.4 | ↓1.9 |
| MoAlign (V → T → N) | 75.6 | 67.3 | 86.1 | ↓2.1 |
| MoAlign (T → N → V) | 75.8 | 67.7 | 85.9 | ↓2.0 |
| MoAlign (T → V → N) | 76.0 | 66.9 | 86.1 | ↓2.1 |

Table 3: Order Impact on FB15K-DB15K (80%).

tributes as "w/o image attribute", our method drops 2.7% and 2.7% on average on FB15K-DB15K and FB15K-YAGO15K. It demonstrates that image attributes can improve model performance and our method utilizes image attributes effectively by capturing more alignment knowledge.

### 4.6 Impact of Multi-modal Attributes

To further investigate the impact of multi-modal attributes on all compared methods, we report the results by deleting different modalities of attributes, as shown in Figure 3. From the figure, we can observe that: 1) The variants without the text or image attributes significantly decline on all evaluation metrics, which demonstrates that the multi-modal attributes are necessary and effective for MMEA. 2) Compared to other baselines, our model derives better results both in the case of using all multi-modal attributes or abandoning some of them. It demonstrates our model makes full use of existing multi-modal attributes, and multi-modal attributes are effective for MMEA. All the observations demonstrate that the effectiveness of the MMKG transformer encoder and the type-prompt encoder.

In addition, to investigate the impact of the order in which multi-modal attributes are introduced to the model, we conduct experiments with different orders of introducing neighbor information, tex-

tual attributes, and visual attributes. As shown in the Table 3, the introduction order has a significant impact on our model. Specifically, the best performance is achieved when textual attributes, visual attributes, and neighbor information are introduced in that order. This suggests that aggregating attribute information to learn a good entity representation first, and then incorporating neighbor information, can effectively utilize both the attributes and neighborhood information of nodes.

### 4.7 Impact of Interference Data

We investigate the impact of interference data on our proposed method for MMEA. Specifically, we randomly replace 5%, 10%, 15%, 20%, 25% and 30% of the neighbor entities and attribute information to test the robustness of our method, as shown in Figure 4. The experimental results demonstrate that our method exhibits better tolerance to interference compared to the baseline methods. This suggests that our approach, which incorporates hierarchical information from different modalities and introduces type prefix to entity representations, is capable of handling interference data and improving the robustness of the model.

### 5 Conclusion

This paper proposes a novel MMEA framework. It incorporates cross-modal alignment knowledge using a two-stage transformer encoder to better capture complex inter-modality dependencies and semantic relationships. It includes a MMKG transformer encoder that uses self-attention mechanisms to establish associations between intra-modal features relevant to the task. Our experiments show that our approach outperforms competitors.

## Limitations

Our work hierarchically introduces neighbor, multimodal attribute, and entity types to enhance the alignment task. Empirical experiments demonstrate that our method effectively integrates heterogeneous information through the multi-modal KG Transformer. However, there are still some limitations of our approach can be summarized as follows:

- Due to the limitation of the existing MMEA datasets, we only experiment on entity, text, and image modalities to explore the influence of multi-modal features. We will study more modalities in future work.

- Our approach employs the transformer encoder architecture, which entails a substantial time overhead. In forthcoming investigations, we intend to explore the feasibility of leveraging prompt-based techniques to mitigate the computational burden and expedite model training.

## Ethics Statement

In this work, we propose a new MMEA framework that hierarchically introduces neighbor, multimodal attribute, and entity types to benchmark our architecture with baseline architectures on the two MNEA datasets.

**Data Bias.** Our framework is tailored for multimodal entity alignment in the general domain. Nonetheless, its efficacy may be compromised when confronted with datasets exhibiting dissimilar distributions or in novel domains, potentially leading to biased outcomes. The experimental results presented in the section are predicated on particular benchmark datasets, which are susceptible to such biases. As a result, it is imperative to exercise caution when assessing the model's generalizability and fairness.

**Computing Cost/Emission.** Our study, which involves the utilization of large-scale language models, entails a substantial computational overhead. We acknowledge that this computational burden has a detrimental environmental impact in terms of carbon emissions. Specifically, our research necessitated a cumulative 588 GPU hours of computation utilizing Tesla V100 GPUs. The total carbon footprint resulting from this computational process is estimated to be 65.27 kg of $CO_2$ per run, with a total of two runs being conducted.

## Acknowledgment

We thank the anonymous reviewers for their insightful comments and suggestions. Jianxin Li is the corresponding author. The authors of this paper were supported by the NSFC through grant No.62225202, 62106059.

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

## A  Related Work

### A.1  Multi-Modal Entity Alignment

In the real-world, due to the multi-modal nature of KGs, there have been several works (Zhu et al., 2022; Wang et al., 2021; Jiang et al., 2021; Fang et al., 2022) that have started to focus on MMEA technology. One popular approach is to use embeddings to represent entities and their associated modalities (Cao et al., 2022; Yang et al., 2022). These embeddings can then be used to measure the similarity between entities and align them across different KGs. Wang et al. (Wang et al., 2018b) proposed a framework that uses cross-modal embeddings to align entities across different modalities, such as text and images. The model maps entities from different modalities into a shared embedding space, where entities that correspond to the same real-world object are close to each other. However, this approach cannot capture the potential interactions among heterogeneous modalities, limiting its capacity for performing accurate entity alignments. To address this limitation, some researchers have proposed multi-modal knowledge embedding methods that can discriminatively generate knowledge representations of different types of knowledge and then integrate them. Chen et al. (2020) proposed a method that uses a multi-modal fusion module to integrate knowledge representations of different types. Similarly, Guo et al. (2021) proposed a GNN-based model that learns to aggregate information from different modalities and propagates it across the knowledge graph to align entities. It developed a hyperbolic multi-modal entity alignment (HEA) approach that combines both attribute and entity representations in the hyperbolic space and uses aggregated embeddings to predict alignments. EVA (Liu et al., 2021) combines images, structures, relations, and attributes information for the MMEA with a learnable weighted attention to learn the importance of each modal attributes. Despite these advances, existing methods often ignore contextual gaps between entity pairs, which may limit the effectiveness of alignment.

### A.2  Knowledge Graph Transformer

The Transformer architecture, originally proposed for natural language processing tasks, has been applied to various knowledge graph tasks as well (Liu et al., 2022; Hu et al., 2022; Howard et al., 2022; Wang et al., 2023, 2022; Fang et al., 2023b,a). For example, the KGAT model (Wang et al., 2019) uses a graph attention mechanism to capture the complex relations between entities in a knowledge graph, and a Transformer to learn representations of the entities for downstream tasks such as link prediction and entity recommendation. The K-BERT model (Liu et al., 2020a) extends this approach by pre-training a Transformer on a large corpus of textual data, and then fine-tuning it on a knowledge graph to improve entity and relation extraction. The transformer model has the ability to model long-range dependencies, where entities and their associated modalities can be distant from each other in the knowledge graph. Furthermore, Transformers utilize attention mechanisms to weigh the importance of different inputs and focus on relevant information, which is particularly useful for aligning entities across different modalities.

## B  Comparision Methods

We compared our method with three EA baselines that aggregate text attributes and relation information, and introduced image attributes initialized by VGG16 for entity representation using the same aggregation method as for text attributes. The three EA methods compared are as follows: (1) **TransE** (Bordes et al., 2013) assumes that the entity embedding $v$ should be close to the attribute embedding $a$ plus their relation $r$. (2) **GCN-align** (Wang et al., 2018a) transfers entities and attributes from each language to a common representation space through GCN. (3) **AttrGNN** (Liu et al., 2020b) divides the KG into multiple subgraphs, effectively modeling various types of attributes.

We also compared our method with three transformer models that incorporate multi-modal information: (4) **BERT** (Devlin et al., 2019) is a pre-trained model to generate representations. (5) **ViT** (Dosovitskiy et al., 2021) is a visual transformer model that partitions an image into patches and feeds them as input sequences. (6) **CLIP** (Radford et al., 2021) is a joint language and vision model architecture that employs contrastive learning.

In addition, we compared our method with four MMEA methods focusing on utilizing multi-modal attributes: (7) **PoE** (Liu et al., 2019) utilizes image features and measures credibility by matching the semantics of entities. (8) **Chen et al.** (Chen et al., 2020) designs a fusion module to integrate multi-modal attributes. (9) **HEA** (Guo et al., 2021) characterizes MMKG in hyperbolic space. (10)

**EVA** (Liu et al., 2021) combines multi-modal attributes and relations with an attention mechanism to learn the importance of modality. (11) **ACK-MMEA** (Liu et al., 2021) designs an attribute-consistent KG representation framework to compensate contextual gaps.