# OpenReview forum: "Multi-Modal Knowledge Graph Transformer Framework for Multi-Modal Entity Alignment"
_EMNLP/2023/Conference — EMNLP 2023 Findings_

### Official Review · Reviewer_Up9k · 2023-07-30

**Typos Grammar Style And Presentation Improvements:** N/A
**Soundness:** 4

**Excitement:**

4: Strong: This paper deepens the understanding of some phenomenon or lowers the barriers to an existing research direction.

**Missing References:**

Vision, Deduction and Alignment: An Empirical Study on Multi-modal Knowledge Graph Alignment.

**Paper Topic And Main Contributions:**

This paper hierarchically introduces neighbor, multimodal attribute, and entity types to enhance the alignment task. The Meaformer framework introduces a novel approach called Multi-Modal KG Transformer (MMKT) to effectively integrate heterogeneous information. It incorporates a hierarchical modifiable self-attention block to establish associations of task-related intra-modal features through layered introduction, while also using an entity-type prefix to enhance entity representations. The proposed framework achieves remarkable performance on public multi-modal entity alignment datasets, outperforming state-of-the-art methods.

**Questions For The Authors:**

No questions.

**Reasons To Accept:**

The strengths of this paper are as follows:
1. Novel MMEA Framework: The paper proposes a novel MMEA framework called "Meaformer," which leverages the multi-modal KG Transformer to effectively integrate heterogeneous information from different types of data in multi-modal knowledge graphs (MMKGs). This framework addresses the challenges posed by the presence of diverse information, such as neighboring entities, multi-modal attributes, and entity types.
2. Hierarchical Modifiable Self-Attention Block: To handle task-related intra-modal features, the paper introduces a hierarchical modifiable self-attention block in the transformer encoder. This design preserves the unique semantics of different information types, enabling better integration of multiple information sources. This attention mechanism likely helps in capturing complex relationships and patterns within the MMKGs.
3. Entity-Type Prefix Injection: The paper devises two entity-type prefix injection methods to incorporate entity-type information using type prefixes. By doing so, it restricts global information sharing for entities not present in the MMKGs. This approach likely improves the alignment task's accuracy by focusing on relevant entity types, enhancing the entity representations.
4. State-of-the-Art Performance: The proposed Meaformer framework demonstrates state-of-the-art performance on public multi-modal entity alignment datasets. This indicates that the approach is highly effective in identifying equivalent entity pairs across MMKGs and outperforms existing strong competitors in the field.
Overall, the paper's strengths lie in its innovative framework design, the use of hierarchical modifiable self-attention, and the effective incorporation of entity-type information, leading to superior performance in multi-modal entity alignment tasks.


**Reasons To Reject:**

Lack of citations and comparisons to recent similar papers in entity alignment. If this latest relevant paper as well as similar papers could be cited, then I think this paper should be accepted as a good paper.

**Reproducibility:**

4: Could mostly reproduce the results, but there may be some variation because of sample variance or minor variations in their interpretation of the protocol or method.

**Reviewer Confidence:**

3: Pretty sure, but there's a chance I missed something. Although I have a good feel for this area in general, I did not carefully check the paper's details, e.g., the math, experimental design, or novelty.

---

> ### Author Rebuttal · Authors · 2023-08-28
>
> **Q1:** Lack of citations and comparisons to recent similar papers in entity alignment. If this latest relevant paper as well as similar papers could be cited, then I think this paper should be accepted as a good paper. \
> **A1:** Thank the reviewer for your high acceptation of our work. LODEME[1] iteratively performs logical deduction and multi-modal embedding and developed a structure-aware attention mechanism such that the entity embeddings can incorporate multiple images with different emphases. Different from the above paper, our paper focuses on designing a Hierarchical Modifiable Self-Attention that hierarchically introduces neighbor features, multi-modal attributes, and entity types to better capture complex interactions between modalities and contextual information. Furthermore, entity alignment works [2-4] designed for the cross-lingual scenario advised by Reviewer adWR. We will add the citations and comparisons to recent similar papers in entity alignment on the related work.
>
> [1] Vision, Deduction and Alignment: An Empirical Study on Multi-modal Knowledge Graph Alignment. Yangning Li, Jiaoyan Chen, Yinghui Li, Yuejia Xiang, Xi Chen, Haitao Zheng, arXiv 2023.\
> [2]Visual Pivoting for (Unsupervised) Entity Alignment, Fangyu Liu, Muhao Chen, Dan Roth, and Nigel Collier, AAAI 2021.\
> [3]Multi-modal Contrastive Representation Learning for Entity Alignment, Zhenxi Lin, Ziheng Zhang, Meng Wang, Yinghui Shi, Xian Wu, and Yefeng Zheng, COLING 2022.\
> [4]MEAformer: Multi-modal Entity Alignment Transformer for Meta Modality Hybrid. Zhuo Chen, Jiaoyan Chen, Wen Zhang, Lingbing Guo, Yin Fang, Yufeng Huang, Yuxia Geng, Jeff Z. Pan, Wenting Song, Huajun Chen, arXiv 2022.

---

### Official Review · Reviewer_adWR · 2023-08-04

**Typos Grammar Style And Presentation Improvements:** 1. In 175-176, "Meaformer also design…
**Soundness:** 3

**Excitement:**

3: Ambivalent: It has merits (e.g., it reports state-of-the-art results, the idea is nice), but there are key weaknesses (e.g., it describes incremental work), and it can significantly benefit from another round of revision. However, I won't object to accepting it if my co-reviewers champion it.

**Missing References:**

The paper misses another method also named "Meaformer" (https://arxiv.org/abs/2212.14454) which was online in December 2022. This works also uses Transformer, and the authors should compare the proposed method to it.

**Paper Topic And Main Contributions:**

This paper presents a Transformer-based method Meaformer for multi-modal knowledge graph entity alignment. It uses BERT and positional encodings to get embeddings, and feed these embeddings into hierarchical modifiable self-attention block.

The evaluation uses two cross-KG EA datasets FB15K-DB15K and FB15K-YAGO15K. The reported performance is better than the adopted baselines.

**Questions For The Authors:**

1. What does "one-dimensional vector" mean in 186? Is it a real?

2. What does the colours of the entities in Figure 1 mean? They could be specified in the caption.

**Reasons To Accept:**

1. A Transformer-based multi-modal knowledge graph entity alignment method is proposed, with promising results achieved.

**Reasons To Reject:**

1. The technical contribution of the proposed method, which uses a Transformer for multi-modal KG EA, is only incremental.

2. The quality of the paper should be improved. Some textual descriptions in this paper are not clear, making it hard to follow. E.g., in 043 - 044, for "due to the heterogeneity of MMKGs (e.g., different neighbors, multi-modal attributes, distinct types)", what does "distinct types" mean? In 230, what does "first-order neighbous" mean? What does "we randomly initialize a reference order" mean? Many texts in this paper are hard to follow.

3. Only two cross MM KG EA benchmarks are tested. There are more cross lingual MM EA benchmarks that can be tested.



**Reproducibility:**

3: Could reproduce the results with some difficulty. The settings of parameters are underspecified or subjectively determined; the training/evaluation data are not widely available.

**Reviewer Confidence:**

3: Pretty sure, but there's a chance I missed something. Although I have a good feel for this area in general, I did not carefully check the paper's details, e.g., the math, experimental design, or novelty.

---

> ### Author Rebuttal · Authors · 2023-08-28
>
> **Q1:** The technical contribution of the proposed method, which uses a Transformer for multi-modal KG EA, is only incremental. \
> **A1:** In this paper, we are not simply employing a vanilla Transformer. On the contrary, we carefully reform the three core components of the Transformer architecture specifically for the MMEA task, including the position encoding, self-attention mechanism, and feed-forward network. First, we introduce the modality and structural information into the position embedding which is critical for the MMEA task. Second, we propose a carefully-designed hierarchical modifiable self-attention mechanism specifically to address the aggregation issue of different information in the MMEA task. Third, we propose a type injection in FFN to enable better filtering out of unsuitable candidates for entity alignment. The above technical contribution lies in the core components (i.e., position encoding, self-attention, and FFN) of the vanilla Transformer architecture. Therefore, we are trying to design a new Transformer for the MMEA task to overcome the aggregation issue, rather than simply using a vanilla Transformer in the task. We thank the reviewer for the question and will refine the contribution part to be clearer.
>
> **Q2:** The quality of the paper should be improved. Some textual descriptions in this paper are not clear, making it hard to follow. \
> **Q2.1:** E.g., in 043 - 044, for “due to the heterogeneity of MMKGs (e.g., different neighbors, multi-modal attributes, distinct types)”, what does “distinct types” mean? \
> **A2.1:** The distinct types mean the entity types. For each entity, we introduce a distinct type. We will change it to distinct entity types for ease of understanding.
>
> **Q2.2:.** In 230, what does “first-order neighbors” mean? \
> **A2.2:** The “first-order neighbors” mean that entities directed connect to the current entity in the multi-modal knowledge graph. We will provide an explanation of this in Section 3.1.2.
>
> **Q2.3:** What does “we randomly initialize a reference order” mean? Many texts in this paper are hard to follow. \
> **A2.3:** For an entity of the MMKG, there are multiple first-order neighbors. For the Transformer architecture, we should input it in a specific order. However, the order is not existing in the MMKG. Thus, we randomly initialize the order. We will give a more detailed explanation.
>
> **Q3:** Only two cross MM KG EA benchmarks are tested. There are more cross lingual MM EA benchmarks that can be tested. \
> **A3:** Thanks for the advice. We supply the experiments on three cross lingual MMEA datasets  $DBP15K_{ZH-EN}$, $DBP15K_{JA-EN}$ and  $DBP15K_{FR-EN}$ with the same experiment settings in the existing papers [1-3]. From the results, our model performs best on MRR and Hits@10. The experimental results demonstrate that our model preserves the unique semantics of different information by introducing a hierarchical modifiable self-attention block, which is suitable for the cross lingual MMEA task. We will add the results in Section 4.
>
> The experiment on  $DBP15K_{ZH-EN}$:
>
> | **Models** | **MRR** | **Hits@1** | **Hits@10** |
> | --- | :---: | :---: | :---: |
> | EVA [1] | 95.1 | 92.9 | 98.6 |
> | MCLEA [2] | 94.6 | 92.6  | 98.3 |
> | MEAformer [3]  | 96.5 | 94.9  | 99.3 |
> | **Ours** | **96.8** | **95.2** | **99.7** |
>
>
> The experiment on $DBP15K_{JA-EN}$:
>
> | **Models** | **MRR** | **Hits@1** | **Hits@10** |
> | --- | :---: | :---: | :---: |
> | EVA [1] | 97.6 | 96.4 | 99.7 |
> | MCLEA [2] | 97.3 | 96.1 | 99.4 |
> | MEAformer [3] | 98.6 | 97.8 |  99.9 |
> | **Ours** | **99.0** | **98.1** | **100.0** |
>
> The experiment on $DBP15K_{FR-EN}$:
>
> | **Models** | **MRR** | **Hits@1** | **Hits@10** |
> | --- | :---: | :---: | :---: |
> | EVA [1] | 99.5 | 99.0 | 99.9 |
> | MCLEA [2] | 99.2 | 98.7 | 99.9 |
> | MEAformer [3] | 99.5 | **99.1** | **100.0** |
> | **Ours** | **99.6** | 98.8 | **100.0** |
>
> [1]Visual Pivoting for (Unsupervised) Entity Alignment, Fangyu Liu, Muhao Chen, Dan Roth, and Nigel Collier, AAAI 2021. \
> [2]Multi-modal Contrastive Representation Learning for Entity Alignment, Zhenxi Lin, Ziheng Zhang, Meng Wang, Yinghui Shi, Xian Wu, and Yefeng Zheng, COLING 2022. \
> [3]MEAformer: Multi-modal Entity Alignment Transformer for Meta Modality Hybrid. Zhuo Chen, Jiaoyan Chen, Wen Zhang, Lingbing Guo, Yin Fang, Yufeng Huang, Yuxia Geng, Jeff Z. Pan, Wenting Song, Huajun Chen, arXiv 2022.
>
> **Q4:** What does “one-dimensional vector” mean in 186? Is it a real? \
> **A4:** Thanks for the comment and it is a writing mistake. We correct the expression as “convert the matrix into a vector similar to word embeddings”.
>
> **Q5:** What does the colours of the entities in Figure 1 mean? They could be specified in the caption. \
> **A5:** The meanings of different colors can already be found in the caption of Figure 1 in the submitted manuscript. We will refine the caption to make it clearer. Specifically, the different colors mean different types of information in an MMKG. The blue ones are entities, the green ones are textual attributes, and the orange ones are visual attributes.
>
> **Q6:** Missing References: The paper misses another method also named “Meaformer”. \
> **A6:** Thanks for your advice. We will add this paper [3] to the related work. It dynamically predicts the mutual correlation coefficients among modalities for entity-level modality fusion. Different from the above paper, our paper focuses on designing an MMKG transformer within only one transformer encoder to maintain the structure information by position embedding and designing a hierarchical modifiable self-attention block.
>
> [3]MEAformer: Multi-modal Entity Alignment Transformer for Meta Modality Hybrid. Zhuo Chen, Jiaoyan Chen, Wen Zhang, Lingbing Guo, Yin Fang, Yufeng Huang, Yuxia Geng, Jeff Z. Pan, Wenting Song, Huajun Chen, arXiv 2022.
>
> **Q7:** In 175-176, “Meaformer also design” –> “Meaformer also designs”. \
> **A7:** We will change the sentence to “Meaformer also designs”.

---

### Official Review · Reviewer_h5aT · 2023-08-04

**Soundness:** 4

**Excitement:**

4: Strong: This paper deepens the understanding of some phenomenon or lowers the barriers to an existing research direction.

**Missing References:**

N/A

**Paper Topic And Main Contributions:**

This paper focuses on multi-modal entity alignment (MMEA). To make better use of different information of MMKGs (neighbors, attributes, and types in this paper), the authors design a prefix-injected MMKG transformer with a hierarchical modifiable self-attention. The experimental results show the effectiveness of the proposed model.

**Questions For The Authors:**

1. How is the problem addressed in this paper different from others?

2. Why does “N-V-T” work better? Is it related to the dataset?

**Reasons To Accept:**

There are three reasons:

A. Improved attention mechanism in KG transformer for MMEA. It is efficient and natural to capture heterogeneous information separately in stages/blocks.

B. The idea of prefix injection is interesting. The type prefixes are injected into both self-attention and FNN, which is helpful for MMEA task. Beyond that, such an idea may inspire others to explore new injection forms of other prefixes in other tasks.

C. SOTA results of sufficient experiments.


**Reasons To Reject:**

There are also three reasons:

A. The problem of “incorporating/aggregating the different types of information” should be stated more clearly, rather than just providing some examples. I believe this is the motivation for designing the new self-attention.

B. As the major contribution of this paper, the order of attention layers is not fully analyzed in experiments. It is recommended to make a supplementary explanation of the reasons for the results of Table 3.

C. In my opinion, the analysis of Related Work is important for this paper. It is recommended to add the corresponding section to the main text.


**Reproducibility:**

4: Could mostly reproduce the results, but there may be some variation because of sample variance or minor variations in their interpretation of the protocol or method.

**Reviewer Confidence:**

5: Positive that my evaluation is correct. I read the paper very carefully and I am very familiar with related work.

**Typos Grammar Style And Presentation Improvements:**

Typos: Line 59, multi-mdoal. Line 176, Meaformer also design. Line 449, Comparision.

Consistency: multi-modal and multimodal, MMKG1 and MMKG 1.

Improvements: Caption of Figure 4, maybe “… means interference *WITH* some proportion of attributes or neighbors.”

---

> ### Author Rebuttal · Authors · 2023-08-28
>
> **Q1:** More clearly description of “incorporating/aggregating the different types of information”. \
> **A1:** Thanks for the encouraging comments. We design the hierarchical modifiable self-attention for the problem of aggregating different types of information. Specifically, we focus on the information of neighbors, multi-modal attributes, and types, which are the major features and structural information used in most works. However, directly aggregating such different information leads to a misaligned information space, as they provide multiple perspectives on entity cognition. The information of neighbors contains the context information of the target entity. The multi-modal attributes incorporate the features unique to the entity itself, which are the internal characteristics of the target entity. The entity type constrains the possible scope of the entity. Different from the previous works, we design a new self-attention mechanism to hierarchically aggregate different information. We will add the description in Section 1.
>
> **Q2:** Make a supplementary explanation of the reasons for the results about the order of attention layers of Table 3. \
> **A2:** Thanks for the encouraging comments. Based on the reviewer’s comments, we added more description and analysis of Table 4. In Table 3, we analyze the effect of the different order of attention layers. Specifically, the best two orders are “N->V->T” and “N->T->V”. It demonstrates that neighbor information provides more personalized information for entity alignment which is consistent with existing works. When the neighbor multi-head attention is introduced in the last, the performance drops dramatically. It demonstrates that early introducing neighbor information can provide more directional information for effectively aggregating textual and visual information. In addition, the order of visual and textual information plays a relatively small role which is likely to be related to the dataset. We will give a supplementary explanation of the reasons for the results of Table 3 in Section 4.6.
>
> **Q3:** Add the Related Work to the main text. \
> **A3:** Thanks for the kind suggestion and we will move the related work from the appendix to the main text.
>
> **Q4:** How is the problem addressed in this paper different from others? \
> **A4:** Existing works mainly focus on aggregating multi-modal information while ignoring whether the information should be aggregated directly, which could lead to incorrect alignment. Different from the previous works, we argue that a differentiated and personalized aggregation of neighbors and modalities should be performed. It helps to prevent the misalignment of cross-modal semantics and the non-discriminative representations of different types of entities.
>
> **Q5:** Why does “N-V-T” work better? Is it related to the dataset? \
> **A5:** We do experiment on another MMEA dataset FB15K-YAGO15K (80%) . The experiment shows that the best order is also“N-V-T”. The experiment also demonstrates that early introducing neighbor information can provide more directional information for the multi-modal entity alignment task. We will add it in Section 4.6.
>
>
> | **Variants** | **MRR** | **Hits@1** | **Hits@10** | **$\Delta$ Avg**  |
> | :---: | :---: | :---: | :---: | :---: |
> | **Meaformer (N->V->T)**  | **76.9** | **68.9** | **88.4**  | - |
> | Meaformer (N->T->V) | 76.5 | 68.3 | 87.8 | 0.5 $\downarrow$ |
> | Meaformer (V->N->T) | 76.1 |  67.7 | 87.5 | 1.0 $\downarrow$ |
> | Meaformer (V->T->N) | 75.0 | 67.1 | 86.7 | 1.8 $\downarrow$ |
> | Meaformer (T->N->V)  | 75.8 | 67.4 | 87.1 | 1.2 $\downarrow$ |
> | Meaformer (T->V->N) | 75.3 | 66.8 | 86.4 | 1.9 $\downarrow$ |
>
>
> **Q6:** Typos: Line 59, multi-mdoal. Line 176, Meaformer also design. Line 449, Comparision. \
> **A6:** Thanks and we will fix the above typos and grammar problems.
>
> **Q7:** Consistency: multi-modal and multimodal, MMKG1 and MMKG 1. \
> **A7:** We will change to multi-modal and MMKG 1 to make consistency.
>
> **Q8:** Improvements: Caption of Figure 4, maybe “… means interference WITH some proportion of attributes or neighbors.” \
> **A8** Yes. We will fix the sentence to “… means interference with some proportion of attributes or neighbors.”.

---

### Meta-Review · Area_Chair_7koX · 2023-09-15

**Recommendation:** 4

**Metareview:**

This paper focuses on multi-modal entity alignment (MMEA). To make better use of different information of MMKGs (neighbors, attributes, and types in this paper), the authors design a prefix-injected MMKG transformer with a hierarchical modifiable self-attention. The evaluation uses two cross-KG EA datasets FB15K-DB15K and FB15K-YAGO15K. The proposed framework achieves remarkable performance on these datasets, outperforming state-of-the-art methods.

---

### Decision · Program_Chairs · 2023-10-07

**Decision:**

Accept-Findings

**Comment:**

This paper focuses on multi-modal entity alignment (MMEA). To make better use of different information of MMKGs (neighbors, attributes, and types in this paper), the authors design a prefix-injected MMKG transformer with a hierarchical modifiable self-attention. The evaluation uses two cross-KG EA datasets FB15K-DB15K and FB15K-YAGO15K. The proposed framework achieves remarkable performance on these datasets, outperforming state-of-the-art methods.